# Clinical Manifestations, Pathogenesis, Diagnosis and Treatment of Peripheral Neuropathies in Connective Tissue Diseases: More Diverse and Frequent in Different Subtypes than Expected

**DOI:** 10.3390/diagnostics11111956

**Published:** 2021-10-21

**Authors:** Lei Jin, Yu Liu

**Affiliations:** Department of Rheumatology and Immunology, Tongren Hospital, Shanghai Jiao Tong University School of Medicine, Shanghai 200336, China; leijin1987@hotmail.com

**Keywords:** peripheral neuropathy, connective tissue disease, pathogenesis, diagnosis, treatment

## Abstract

Purpose of review: To discuss and summarize recent findings in peripheral neuropathy (PN) related to connective tissue diseases (CTD) including its prevalence, clinical manifestations, pathogenesis, diagnosis and treatment. Recent findings: Although PN is a common complication in CTD and has been well studied, recent research has shown that PN is more diverse and frequent in different subtypes of CTD than was expected. The incidence of PN in Sjögren’s syndrome and rheumatoid arthritis (RA) varies according to different disease subtypes, and the pathogenesis of neuropathic pain in different subtypes of eosinophilic granulomatosis with polyangiitis (EGPA) may also differ. Neurogenic inflammation, autoantibody-mediated changes, ischemia of the vascular wall and metabolic mechanisms have been shown to contribute to the pathogenesis of PN in CTD. Moreover, allergic inflammation has been recently identified as a possible new mechanism producing peripheral neuropathic pain associated with MPO-ANCA negative EGPA patients. Glucocorticoids are routinely used to relieve pain caused by PN. However, these steroids may cause hyperalgesia, exacerbate neuropathic pain, and activate the early phase of pain induction and produce hyperalgesia. Recently, neuroactive steroids, such as progesterone, tetrahydroprogesterone and testosterone, have been shown to exert protective effects for several PN symptoms, and in particular neuropathic pain. Neuroactive steroids will be an interesting topic for future research into PN in CTD. Summary: It is essential for the diagnosis and treatment of PN in CTD to be updated. Timely diagnosis, appropriate treatments, and multidisciplinary care are essential to minimize morbidity and decrease the risk of permanent neurologic deficits. Further studies are needed to guide diagnosis and treatment.

## 1. Introduction

Connective tissue diseases (CTD) are chronic inflammatory autoimmune diseases induced by antibodies or T-cell responses directed against self-antigens, which can affect all body systems, including the central nervous system (CNS) and peripheral nervous system (PNS) [1]. When the PNS is involved in CTD, peripheral neuropathy (PN) is the most common complication [2], which comprises a heterogeneous group of disorders, such as mononeuropathy, polyneuropathy and mononeuritis multiplex. PN may be a manifestation or a characteristic sign of immune system dysfunction, with variable prevalence and prognosis in CTD. Therefore, rapid recognition and treatment are essential. However, due to a varied complex spectrum of overlapping clinical manifestations, PN is an under-diagnosed complication in CTD and a particular challenge for rheumatologists and neurologists. Glucocorticoids and immunosuppressants are usually administered as basic and routine treatments of PN in CTD. However, as reported in experimental models of neuropathic pain, glucocorticoids may cause hyperalgesia, exacerbate neuropathic pain, and activate the early phase of pain induction and indeed produce hyperalgesia [3]. A possible strategy to find an effective treatment for PN is shifting the focus to new biological targets and relevant molecular events in the PNS; in particular, neuroactive steroids are a highly promising therapeutic option [4] as these steroids can modulate PNS functions. This review will discuss and summarize the latest knowledge about PN related to CTD in relation to the following aspects, namely its pathogenesis, clinical manifestations, diagnosis and treatment.

## 2. Methods

A literature review of original articles, review articles and case reports was conducted using Pubmed, Embase and Cochrane databases from January 2015 to June 2021. The search terms were: peripheral neuropathy; systemic lupus erythematosus; Sjögren’s syndrome; systemic sclerosis; mixed connective tissue disease; dermatomyositis; polymyositis; systemic vasculitis; Behçet disease; rheumatoid arthritis; anti-TNF-α therapy; and neuroactive steroids.

Findings of literature review

Classification

PN can be classified into multiple descriptive types according to different criteria (Figure 1) [5]. The most applied criterion of PN classification by neuropathologists is anatomical structure, in terms of the peripheral nerves affected by PN. Therefore, we describe the anatomical structure and classification of the peripheral nerve to facilitate further understanding of the PN classification.

2. Prevalence and clinical manifestations

PN is a general term for disease that involves structural or functional impairment of the PNS, with various causes such as infection, autoimmune disorders (e.g., rheumatoid arthritis, Sjögren’s syndrome and systemic lupus erythematosus), systemic or metabolic disorders, after exposure to toxic compounds and during drug therapy [4]. Table 1 summarizes the main electrodiagnostic test patterns and the main forms of PN involved in several common CTD. The classification of PN as demyelinating or axonal lesions provides a more accurate etiological approach. Demyelinating neuropathy may be inherited or acquired, while symmetrical axonal sensorimotor polyneuropathy is mainly related to autoimmune, endocrine or metabolic diseases.

Axonal sensory polyneuropathy and sensorimotor polyneuropathy can be characterized by paresthesia and defects (including mild touch, proprioception and vibration sensation) in the distal part of the symmetrical limb, mainly affecting the distal end of the lower limbs, and may be accompanied by burning pain in the feet. In addition to the above manifestations, motor weakness may be present in sensorimotor polyneuropathy, which is usually mild and limited to the extensor muscles of the toes or feet [5].

Small-fiber neuropathy (SFN) is an algetic esthesioneurosis that usually results in burning pain and arises in the early stage of several systemic diseases such as diabetes, amyloidosis and CTDs [6]. The main manifestations of small fiber neuropathy are numbness, burning sensation, electric pain, pricking, pruritus, involving the limbs, trunk or the proximal part of the face [5]. Motor neuron disease is characterised as paresis, atrophy and bundle fibrillation, mainly in the distal limb [5]. Besides, SFN consists of two different types, which may be underestimated. The first is called “length-dependent” SFN, which is a neuropathic pain arising in a distal “stocking-and-glove” distribution reported by the patients. The conventional model of this distal neuropathic pain is related to equivalent skin biopsy markers of the most distal axonal degeneration. These markers include reduced intra-epidermal nerve-fiber density (IENFD) of amyelinic nerves. Compared to the proximal leg, the fiber density is decreased at the distal leg. While, concerning the second type of the disease which is called “non-length-dependent” SFN, patients suffer from heterodox and atypical models of neuropathic pain which involves the face, truncus and proximal arms and legs. However, skin biopsy results reveal that this non-conventional model of proximal neuropathic pain is related to skin biopsy markers which show that neuronal degeneration affects the most proximal component in the PNS—the dorsal root ganglia (DRG). Under the circumstances, the IENFD in the distal leg is not decreased any more compared to that in the proximal leg [6].

**Figure 1 diagnostics-11-01956-f001:**
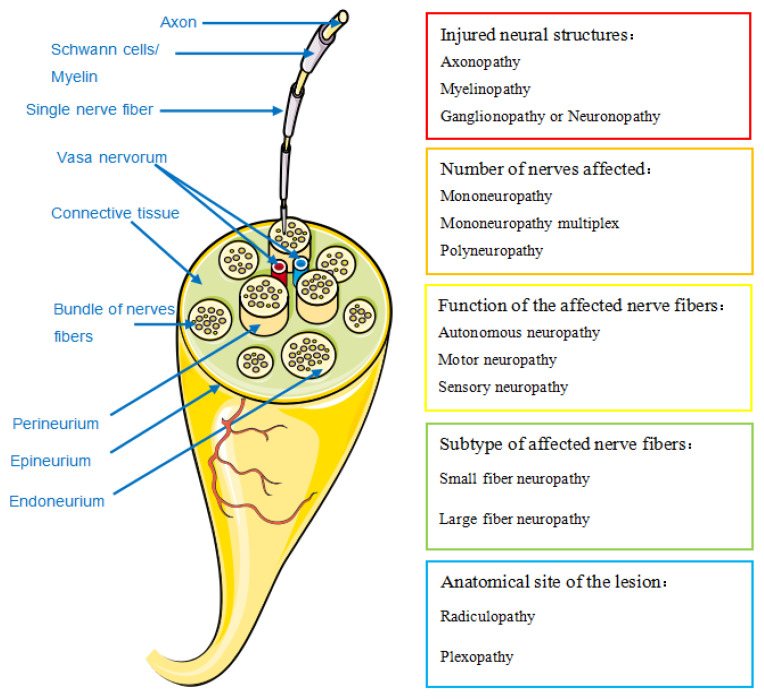
Anatomy of peripheral nerve and classification options for peripheral neuropathy. According to the structures, number, function, subtype, anatomical site of the lesion of the affected nerves, peripheral neuropathy can be classified into different categories [5].

As a heterogeneous group of neurological disorders, the reported prevalence and clinical manifestations of PN in CTD varies widely (Table 1). As shown in Table 1, the main studies during the last 5 years have focused on systemic lupus erythematosus (SLE), Sjögren’s syndrome (SS), systemic sclerosis (SSc), antineutrophil cytoplasmic antibody (ANCA)-associated vasculitis and rheumatoid arthritis (RA). It is still worth mentioning that up to one-third of PN cases have a non-CTD aetiology including infection, drug toxicity or metabolic diseases. The final attribution of PNS involvement in CTD is therefore a relevant and challenging clinical issue [7,8].

### 2.1. Systemic Lupus Erythematosus (SLE)

Among the studies that have investigated the prevalence of PN in autoimmune diseases in recent years, SLE was one of the major diseases studied. In 2019, Shaban et al. [9] reported that the prevalence of PN-SLE was 3.4–7.5% in their research review. Moreover, cross-sectional studies in recent years have revealed that PN is a very common complication in SLE, with the prevalence ranged from 1.5% to 36%, as highlighted in Table 1 [8,10,11,12,13,14], and which was positively related to the level of disease activity [8,11,12,13,14]. The large degree of variation in PN prevalence in these studies was mainly attributed to the sample size, the different features of each group such as drug use, disease activity and/or ethnicity. PN symptoms were usually observed in the first 5 to 7 years of SLE being diagnosed [10,14], but PN events were a component of the first symptoms once SLE had been diagnosed [13]. Neurodiagnostic analysis revealed a predominance of sensory-motor involvement and axonal patterns [10,11,13,14]. Although polyneuropathy was the main form of PN manifestation, about half of the polyneuropathy events were attributed to non-SLE, mononeuritis multiplex and cranial neuropathies likely related to SLE [12]. The most common nerves affected by PN in SLE were the peroneal nerves, followed by the tibial and sural nerves, while ulnar and median nerves were less affected [10]. The most frequent cranial neuropathies were II, followed by VIII, VII, V, VI, IX, III, I and IV [13].

Although Guillain–Barré Syndrome (GBS) is a rare manifestation of SLE, which falls under the category of PNS-SLE, it was not mentioned in these studies. It is, however, important to remember that GBS can be one of the main causes of morbidity and mortality in PN-SLE patients [15,16].

In conclusion, SLE patients are easily susceptible to PN, but neurodiagnostic analysis of PN-SLE varies widely. PN-SLE should be given greater recognition. Further studies should focus on the differences between individual subtypes of neuropathy to explore the different pathologies and guide diagnosis and treatment.

### 2.2. Sjögren’s Syndrome (SS)

SS is the second most common chronic autoimmune rheumatic disease, including primary Sjögren’s syndrome (pSS) and secondary Sjögren’s syndrome (sSS). We found six valid cross-sectional studies and one cohort study that discussed the prevalence and clinical features of PN in SS [17,18,19,20,21,22,23]. In the studies on pSS, the reported prevalence of PN events ranged from 19% to 72% and could be the initial manifestation [17,18,19,20,21,22,23]. Only one cross-sectional study from China in 2018 compared pSS with sSS and reported that sSS patients had a higher prevalence of PN events than pSS patients (31.1% vs. 19%) [17]. Several previous studies found a higher frequency of symmetric sensorimotor polyneuropathy and symmetric sensory polyneuropathy [24,25], but mononeuropathy or mononeuritis multiplex was the most common pattern in these studies [21,22,23]. It is difficult to estimate the precise determination of the prevalence of these manifestations, partly due to the criteria-related variations for inclusion of patients and those related to the recruitment of patients studied, and the manifestations considered. However, in addition to the above two forms of PN manifestations, cranial neuropathy (mainly trigeminal neuropathy) and entrapment neuropathy (mainly carpal tunnel syndrome) are not rare. An acute or a subacute onset was observed more frequently for multiple cranial neuropathies. Atypical presentations included pure motor neuropathies [18], hypertrophic neuropathy [26] and ganglionopathy [21].

In conclusion, PN events are common in SS. Further studies should focus on the differences in the incidence and clinical patterns of PN events between sSS and pSS. PN might help to establish new international classification criteria and clinical practice guidelines for pSS in the future.

### 2.3. Systemic Sclerosis (SSc) or Scleroderma

There have been few studies on PN-SSc in recent years, with varying sample sizes and definitions for PN events (some studies used questionnaires to define neuropathic pain). Raja et al. [27] and Paik et al. [28] performed cross-sectional studies and reported that the prevalence of PN-SSc varied from 28% to 36.6%. A systematic review from Turkey concluded that trigeminal neuropathy (TN) (16.5%), peripheral sensorimotor polyneuropathy (14.3%), and carpal tunnel syndrome (CTS) (6.6%) were the most frequent forms of PN-SSc [29], and individual cases of TN in SSc were mainly reported in the last 5 years. In the studies by Yagci et al. [29] and Sriwong PT et al. [30], the prevalence of median neuropathy in SSc was about 35%. Most CTS in patients with SSc were asymptomatic. Autonomic nervous system (ANS) dysfunction, especially cardiac autonomic functions may occur in SSc [31,32], leading to an increased sympathetic modulation and decreased vagal at rest and a blunted autonomic response to orthostatism [31,32]. Most of these changes were detectable in the advanced and fibrotic forms of SSc. [33]. More well-designed studies are still needed.

### 2.4. Polyarteritis Nodosa (PAN)

One small sample size cross-sectional study from India calculated the prevalence of PN in PAN [34]. The prevalence was 88.9% (22 of 27 cases), the main form of which was axonal injured-sensorimotor mononeuritis multiplex [34]. De Boysson et al. [35] and Imboden et al. [36] mentioned that between 65% and 85% of PAN patients presented with PNS disorders. Polyneuropathy, radiculopathies, lumbar or brachial plexopathies have been reported. Acute neuropathy can also occur in necrotizing vasculitis, but very few cases have been reported [37].

In summary, among CTD, PNA has a high prevalence of PN. The mechanism is arteritis of the vasa nervorum, leading to ischemic neuropathy [36]. Mononeuritis multiplex is the main form of PN manifestation, probably because PAN mainly affects the moderate vessels, which supply slightly larger nerves.

### 2.5. Antineutrophil Cytoplasmic Antibody (ANCA)-Associated Vasculitis

In 2019, Bischof et al. [38] carried out a large cross-sectional study. Nine hundred and fifty-five patients were identified as ANCA-associated vasculitis (AAV), among which 572 were granulomatosis polyangiitis (GPA), 218 microscopic polyangiitis (MPA) and 165 EGPA. The prevalence of PN involvement was 65% in EGPA, 23% in MPA and 19% in GPA. In another two retrospective cohort studies, Zhang et al. [39] and Cho et al. [40] reported that the prevalence of PN events in EGPA was between 46.4% and 75%. Unlike PAN, PN-EGPA patients were characterized by polyneuropathy. Some researchers speculated the reason is that EGPA primarily affects small vessels which supply terminal nerves but PAN mainly affects the moderate vessels, which supply slightly larger nerves. Nishi et al. [41] compared sural nerve biopsy specimens of 27 PN-EGPA patients with positive MPO-ANCA and 55 PN-EGPA patient specimens with negative MPO-ANCA, and found that the MPO-ANCA positive group was mainly characterized by vasculitis in epineurial vessels, while the MPO-ANCA negative group was mainly characterized by eosinophil infiltration, suggesting that the pathogenesis of PN-EGPA comprises at least 2 distinct mechanisms (possible mechanisms are briefly described in Section 3.2 and Section 3.3 below). Further larger scale studies are needed to clarify the clinical and pathological relationships between ANCA positivity and PN involvement in EGPA patients.

PN in EGPA primarily affected the lower extremities, with peroneal nerve involvement being the most frequent and severe. The sensory neuropathy was distributed mostly asymmetrically in the distal portion of the limbs, while the main manifestations of motor neuropathy were foot drop and muscle weakness. AVV patients may also have symptoms of autonomic dysfunction which is independent of the disease duration and its severity [42]. Atypical manifestations include acute sciatic nerve neuropathy [43] and mimicking GBS [44]. 

Mononeuritis multiplex and distal symmetrical polyneuropathy, mixed neuropathy and lower limbs involvement are predominant PN manifestations in MPA patients [45,46]. Symmetrical sensorimotor polyneuropathy is rare in GPA [47], with only one case report in the last 5 years, characterized by CTS and tennis elbow as prodromes [48].

### 2.6. Rheumatoid Arthritis (RA)

PN in RA was mainly reported as small sample sizes in cross-sectional studies in the past 5 years. Kaeley et al. [49] reported that the prevalence of PN-RA was 75.3%. Interestingly, the incidence of PN varied according to the different subtypes of RA. Kumar et al. [50] reported that the prevalence of PN in seropositive RA patients and seronegative RA patients was 34.4% vs. 15.4, respectively. Further studies are required to clarify the clinical and pathological relationships between seropositive and seronegative RA patients.

Previous studies have shown that the types of PN-RA are pure sensory, distal axonal sensory-motor, mononeuritis multiplex and entrapment neuropathy [51]. However, Kaeley et al. [49] found that pure motor neuropathy was not rare. Autonomic dysfunction also occurred in RA, characterized by heart rate responses to a deep breath (HRD), heart rate response to standing (HRS), blood pressure response to hand grip and sudomotor function impairment [52]. Rare but serious peripheral nerve manifestations included ischemic neuropathies caused by necrotizing arteritis of the vasa vasorum [53].

### 2.7. Other CTDs

#### 2.7.1. Giant Cell Arteritis (GCA) and Takayasu Arteritis (TAK)

Studies on PN in GCA and TAK were mainly reported as individual cases during the past 5 years. A literature review by Bougea et al. [47] found that PN complications affected 15% of GCA patients. The main neuropathic complication of GCA was CTS, and mononeuritis multiplex and distal symmetrical sensorimotor polyneuropathy were uncommon [47]. Bilateral acute brachial radiculoplexopathy is a rare PN event in GCA, which presents as a “man-in-the-barrel” syndrome (diplegia of the upper extremities in which mobility of the head and lower limbs is preserved) [54,55]. Duval [54] and Calle-Lopez [55] reported this rare neurological complication of GCA, suggesting that GCA should be considered in patients over 50 years old who manifest with peripheral nerve clinical features such as brachial diplegia, and without other demonstrable causes. Also some cases with atypical symptoms such as compressive common peroneal neuropathies [56] have been reported. 

PN involvement of TAK is much rarer. Isolated cranial nerve palsies was reported to be due to the involvement of the internal carotid artery or its branches [47]. Vasculitic neuropathy in TAK may cause a subacute sensorimotor deficit in a cervicobrachial plexus distribution [47]. Although TAK rarely affected the axillary artery, Kim et al. [57] reported a case of a right axillary artery aneurysm in a young female TA patient, which led to brachial plexus injury and compression, causing neurological complications. 

In summary, PN in systemic vasculitis also requires a consideration of non-vasculitic neuropathies factors, especially compression neuropathies.

#### 2.7.2. Behçet Syndrome (BD)

PNS involvement in BD is extremely rare, only isolated cases of distal symmetrical polyneuropathy and mononeuritis multiplex have been reported [47]. A retrospective study from Korea in 2015 reported the overall prevalence of CTS among BD patients was 0.8% [58]. The nerve dysfunction or PN in BD is an axonal type of distal polyneuropathy and predominantly involves the lower extremities [51].

#### 2.7.3. Mixed Connective Tissue Disease (MCTD)

Previous studies suggested that the prevalence of PN-MCTD is approximately 10% to 17% [59], among which trigeminal neuralgia is often associated with MCTD [60]. Bilateral facial nerve palsy with facial swelling can also present in MCTD, and is known as Melkersson–Rosenthal syndrome [61]. Vasculitic neuropathy may be concomitant and manifest as distal symmetric neuropathy [62]. Compressive neuropathies such as CTS can also be observed [62].

#### 2.7.4. Dermatomyositis (DM) and Polymyositis (PM)

Previous studies showed a prevalence of 7.5% in DM/PM patients with polyneuropathy [46]. In 2016, Irie et al. [63] analyzed 9 cases of PN-DM/PM and confirmed that the main form of PN-DM/PM manifestation was axonal neuropathy. Unfortunately, there were no well-designed studies on neuromyositis.

#### 2.7.5. IgG4-Related Disease (IgG4-RD)

IgG4-RD can affect a wide range of visceral nerves including those innervating the kidney, prostate gland, epicardium, abdominal aorta, retroperitoneum and mesentery. The nerves comprising Auerbach’s plexus of the intestinal wall were most extensively involved [64]. This might be one mechanism of the intense pain experienced by IgG4-related retroperitoneal fibrosis patients. The involvement of extremities nerves has only rarely been reported [65].

### 2.8. Peripheral Neuropathies Caused by Immunotherapy

Janus kinase (JAK) inhibitors have been used to treat RA and other CTDs, basically targeting different molecules in the same signal pathway [66]. In a study of JAK1 and JAK2 inhibitor treatment of myelofibrosis, new-onset treatment-related peripheral neuropathy was observed in 22% of patients (sensory symptoms) [67]. Whether the side effects such as PN will occur in the JAK inhibitors treatment of RA needs further study.

PNS involvement in ankylosing spondylitis (AS) and psoriatic arthritis (Pisa) were mainly related to tumor necrosis factor-α (TNF-α) antagonists, such as infliximab, adalimumab and etanercept [68]. In 2015, Tsouni et al. [69] studied the clinical, electrophysiological and frequency of anti-TNF-α (α-TNF) medication-induced neuropathies (ATIN) in patients with inflammatory disorders. Of 2017 patients treated with α-TNF medication, 12 were diagnosed as ATIN with a prevalence of 0.60% and an incidence of 0.4 cases per 1000 person-years. Six patients had focal or multifocal peripheral neuropathies. During the last 5 years, a number of cases of axonal neuropathy [70] or multifocal-motor-neuropathy-like disease [71] associated with the use of infliximab have been reported. Some TNF-α antagonists have been associated with the occurrence of GBS [72]. Patients treated with TNF-α antagonists can develop a GBS-like disease within the first 6 months after the start of therapy, and the symptoms can persist for up to 2 years [5].

**Table 1 diagnostics-11-01956-t001:** Prevalence and clinical manifestation of PN in patients with CTD.

Authors	Prevalence/ Constituent Ratio (%)	Patients (N)	Type of Study	Main Electrodiagnostic Tests Pattern	Main Form of PN Manifestation
Systemic lupus erythematosus					
Xianbin et al. [8]	1.5%	4924	Cross-sectional	Sensory (67.5%), motor (49.3%)	Polyneuropathy +++ Mononeuropathy ++ Cranial neuropathy ++ Myasthenia gravis ++
Toledano et al. [10]	17.7%	524	Cross-sectional	Sensory-motor (56%), axonal 80.3%	Polyneuropathy +++ Mononeuropathy ++ Cranial neuropathy +
Saigal et al. [11]	36%	50	Cross-sectional	Sensory-motor, axonal	-
Bortoluzzi et al. [12]	6.9%	1224	Cross-sectional	Sensory-motor (25%)	Polyneuropathy +++ Cranial neuropathy +++ Mononeuropathy ++ Mononeuritis multiplex +
Hanly et al. [13]	7.6%	1827	Cohort	Sensory-motor (71%), sensory (16.1%) axonal (41.7%), demyelination (21.7%)	Polyneuropathy +++ Mononeuropathy ++ Cranial neuropathy ++ Mononeuritis multiplex ++
Fargetti et al. [14]	1.8%	2074	Cohort	Sensory-motor (68.4%), axonal (49.3%)	Polyneuropathy +++ Mononeuropathy ++ Polyradiculoneuropathy + Cranial neuropathy +
Sjögren’s syndrome					
Ye W et al. [17]	19% pSS 31.1% sSS	415 pSS 151 sSS	Cross-sectional	-	-
Seeliger et al. [18]	44 SS + PNP	108 PNP	Cross-sectional +case-control	Motor (100%), sensory (89%) axonal (36%), demyelinating (23%), both (41%)	-
Carvajal Alegria et al. [19]	16%	392	Cohort	Sensory (57%), sensory-motor (33%)	Mononeuritis multiplex Polyneuropathy Cranial neuropathy
Przyńska-Mazan et al. [20]	63.9%	61 pSS	Cross-sectional	Sensory-motor axonal (47.5%), demyelination, both (5.1%)	Polyneuropathy +++ Mononeuropathy +++ Entrapment neuropathy ++ Mononeuritis multiplex ++
Sireesha et al. [21]	-	20 pSS 1 sSS	Cross-sectional	-	Mononeuritis multiplex +++ Ganglionopathy ++ Trigeminal neuropathy ++
Jaskólska et al. [22]	72%	50 pSS	Cross-sectional	Sensory-motor axonal (22%)	Entrapment neuropathy +++ Mononeuropathy ++ Cranial neuropathy +
Jaskólska et al. [23]	46%	50 pSS	Cross-sectional	Sensory-motor (47%)	Mononeuropathy ++ Cranial neuropathy ++
Systemic sclerosis (scleroderma)					
Raja et al. [27]	36.6%	60	Cross-sectional	Sensory (65%), motor (53%)	Polyneuropathy +++ Mononeuropathy ++ Entrapment neuropathy ++
Paik et al. [28]	28%	60	Cross-sectional	Sensory-motor axonal, no demyelinating	-
* Yagci et al. [29]	29.2%	24	Cross-sectional	-	Entrapment neuropathy Polyneuropathy
* Sriwong et al. [30]	38%	50	Cohort	-	Median neuropathy at the wrist
Polyarteritis nodosa					
Sharma et al. [34]	88.9%	27	Cross-sectional	Axonal sensory-motor (81.8%)	Mononeuritis multiplex
Eosinophilic granulomatosis with polyangiitis					
Bischof et al. [38]	19% 23% 65%	572 GPA 218 MPA 165 EPGA	Cross-sectional	Sensory-motor (32%), sensory (16%), motor (5%)	Mononeuritis multiplex +++
Zhang et al. [39]	46.4%	110 EPGA	Retrospective cohort	-	Polyneuropathy +++ Mononeuritis multiplex ++
Cho et al. [40]	75%	61 EPGA	Retrospective cohort	Sensory (44/46), motor (24/46)	Mononeuritis multiplex +++ Mononeuropathy ++ Polyneuropathy ++
Nishi et al. [41]	-	82 EPGA	Retrospective	Axonal	-
Rheumatoid Arthritis					
Kaeley et al. [49]	75.28%	89	Cross-sectional	Asymmetrical sensorimotor axonal neuropathy, pure motor	Mononeuritis multiplex Entrapment neuropathy
Kumar et al. [50]	34.4% (seropositive) 15.38% (seronegative)	60	Cross-sectional	-	-

* Studies focused on carpal tunnel syndrome. PN, peripheral neuropathy; pSS, primary Sjögren’s syndrome; sSS, secondary Sjögren’s syndrome; PNP, polyneuropathy; GPA, granulomatosis with polyangiitis; MPA, microscopic polyangiitis; EGPA, eosinophilic granulomatosis with polyangiitis; Prevalence/Constituent ratio: +: <10%; ++: 10–30%; +++: >30%.

## 3. Pathogenesis

In CTD, neurogenic inflammation, autoantibodies-mediated changes, ischemia of the vascular wall and metabolic mechanisms are believed to contribute to the pathogenesis of PN and to be predominant in different diseases. Despite significant advances in our understanding the pathogenesis, no single pathogenetic mechanism is thought to be responsible for PN in CTD. Next, we discuss the possible mechanisms underlying disease pathogenesis. 

### 3.1. Peripheral Neuropathies Associated with Neurogenic Inflammation

An increase in proinflammatory cytokine concentrations has been found in patients with vascular neuropathy [7]. Nociceptors located in nerve endings can sense IL-1β and TNF-α directly and induce activation of MAP kinases, resulting in increased membrane excitability. In addition, MAP activation leads to the release of different neuropeptides such as calcitonin gene-related protein, substance P, nitric oxide and chemokines, which subsequently cause vasodilatation, increases in vascular permeability and cell trafficking [7] (Figure 2A). On the other hand, these mediators released from sensory neurons in the periphery directly attract and activate immune innate cells and adaptative immune cells such as T lymphocytes [7]. Nerve growth factor and prostaglandin E2 are major inflammatory mediators released from immune cells that act on sensory neurons inducing peripheral sensitization and hyperalgesic phenomena [7] (Figure 2B).

### 3.2. Hypothetical Mechanisms Underlying Allergic Inflammation-Related Neuropathic Pain (NeP)

The sural nerve biopsy specimens of PN-EGPA patients with negative MPO-ANCA were mainly characterized by eosinophil infiltration [41], suggesting that allergic inflammation was an underlying mechanism in MPO-ANCA negative EGPA patients. Through animal experiments, Fujii et al. [73] found that peripheral nerve damage caused by allergic inflammation can induce Nep. As for allergic individuals, increased humoral immunity may lead to anti-plexin D1 antibody production via molecular mimicry with environmental allergens. Anti-plexin D1 antibodies can damage primary pain-conducting neurons, thus inducing neuropathic pain [73]. In addition, the overproduction of ET-1 in inflamed skin tissues and sera may induce blood–brain barrier (BBB) hyperpermeability and activate microglia and astroglia through the ET-1/EDNRB pathway in allergic inflammation, thus causing NeP [73] (Figure 2C).

### 3.3. Vasculitic Neuropathy

The most possible pathogenesis of vasculitis-related PN is inflammation of precapillary arteries in the nerves [74]. Deposition of immune complexes or T cell-mediated immunity plays a major role in inducing the immunological inflammation and necrosis of vessel wall [74] (Figure 2D). The end result of both processes is the induction of immunological inflammation and necrosis of blood vessel walls, which eventually leads in addition to focal or multifocal, axonal, ischemic neuropathy [74].

### 3.4. Nodes of Ranvier and Autoantibodies

A process of molecular mimicry may act as the starting motif to target different specific antigens within the structure of a nerve [7]. Nodes of Ranvier may be a vulnerable target for autoimmunity due to the intrinsically elevated number of potential antigens and the crucial permeability of the blood–nerve barrier in nodal and juxtaparanodal structures [7] (Figure 2E).

IgG and IgM anticardiolipin antibodies were detected in the serum of CTD patients [75]. The existence of anti-ganglioside antibodies in PN-SLE patients has been found frequently [7], while the chronic inflammatory demyelinating polyneuropathy (CIDP) associated with IgG4 antibodies to neurofascin-155 (NF155) was recently described [76]. These immune antibody markers have been not only proven to be useful in clinical practice but also uncovered novel pathophysiological mechanisms, clinical phenotypes, therapeutic responses and prognosis indicators.

### 3.5. Metabolic Disorders

PNS involvement in CTD can also be caused by metabolic disorders secondary to aggressive therapy, multiorgan pathology and endocrine abnormalities. Metabolic disorders may induce a reaction of demyelinating neuropathy and axon dystrophy in severe cases [75].

## 4. Diagnosis

The diagnosis of PN-CTD is based on recognition of the clinical symptoms and is supported by nerve conduction studies and neuropathology. A careful review of the illness history, current medication and laboratory results will help to arrive at a correct diagnosis.

### 4.1. Nerve Conduction Studies (NCS)

NCS can reveal the asymmetric or multifocal nature of neuropathy. Electromyogram (EMG)/NCS can establish whether patients have sensory-motor neuropathy, sensory neuronopathy or motor neuronopathy. Moreover, EMG/NCS can be applied to distinct polyradiculopathy with mononeuritis multiplex, as both patterns have very similar diffuse, non-length dependent neuropathies. Importantly, EMG/NCS can also be used to identify whether the affected structure of nerves is the axon itself or demyelination. In addition, needle electromyography is very useful in estimating the disease course, the extent of the injury and the existence of a superimposed myopathy [77].

### 4.2. Laboratory Predictors

In Table 2, we summarized the factors related to PN in CTD patients. Apart from neurological examinations (vide supra), the following laboratory indicators can be used to evaluate and predict PNS in CTD.

In SLE, Saigal et al. [11] reported that PN-SLE patients had a statistically significant higher occurrence of pyuria, pleuritis and leukopenia. Xianbin et al. [8] reported that complement 3 (C3) hypocomplementemia and anti-Sm antibody (anti-Sm) positivity occurred more frequently, and remarkably elevated concentrations of serum immunoglobulin G (IgG) were measured in PN-SLE patients compared to non-PN-SLE patients. Ye et al. [17] reported low complement (C3) concentrations, xerophthalmia, ANA positivity, cardiac involvement and that labial salivary gland histological results were good ways to predict PN in SS. In AVV, an increased IgG4 concentration in serum and infiltration of IgG4-positive plasma cells were reported in AVV patients with PN [79], and the authors speculated that epineurial IgG4-positive plasma cell infiltration was correlated with the extent of epineurial fibrosis. Thrombocytosis may be one of the risk factors for PN-MPA [45]. In RA, Li et al. [78] reported that a decreased blood albumin concentration was a probable risk factor for PN-RA. Yesil H et al. [80] found that vitamin D deficiency was also associated with NeP in RA patients.

### 4.3. Histopathological Techniques

The sural nerve is most frequently used for biopsy, which contributes to the determination of the nature and extent of PN [5]. Bischof et al. [38] performed nerve biopsies in 31 PN-AVV patients and showed that 55% of patients had definite vasculitis. Similarly, about 80% of PN-EGPA patients had extravascular eosinophils and 77% of the patients had vasculitis, while no extravascular granuloma was observed in the total of 44 patients [40]. 

In contrast, skin biopsy is an emerging, minimally invasive method for differentiating between length-dependent and non-length-dependent small fiber neuropathy [81], could aid the etiological differential diagnosis [82] and be applied to identify small fiber involvement in mixed neuropathies and for follow-up studies [82]. Lastly, skin biopsy let to study, over the epidermal nerve fibers, also the autonomic fibers that innervate sweat gland, pilomotor muscle and hair follicles, which widen the clinical spectrum of peripheral neuropathy and provide new insight into the pathophysiological mechanisms [83]. The standard biopsy site is the calf, 10 cm above the lateral malleolus. The interfascicular nerve bundles that consist of large myelinated fibers should also be examined in a muscle biopsy with the neuromuscular junctions [77].

### 4.4. Diagnosis of Small Fibers Neuropathy

Somatic and autonomic functional nervous evaluation of SFN involves the sympathetic and parasympathetic autonomic functions, which is realized by determining the psychophysical sensory thresholds (e.g., cold and heat) through quantitative sensory testing (QST), pain-related tests and recording of laser-evoked potentials (LEP), single axon recording utilizing microneurography and tests [84]. On the other hand, in 2010, the European Federation of Neurological Societies and the Peripheral Nerve Society joint amended the Guidelines on the Application of Skin Biopsy in the Diagnosis of PN, and a conclusion was made that distal leg skin biopsy with quantification of the linear density of IENFD, adopting universally accepted counting rules, is a reliable and efficient technology to evaluate the diagnosis of SFN [85]. In fact, the process toward the determination of the diagnosis of SFN in individual patients, beginning from the chief complaints of sensory symptoms, is on the basis of the clues from skin biopsy and/or QST results. The combination of clinical signs and abnormal QST and/or IENFD findings can reliably be used to diagnose SFN compared with the combination of abnormal QST and IENFD findings without clinical signs [86].

What is more, current diagnostic technologies for SFN are also composed of quantitative sensory testing with determination of warm and cold detection thresholds (WDT, CDT), recording of LEP and sympathetic skin responses (SSRs), and measurement of electrochemical skin conductance (ESC) utilizing Sudoscan(^®^) device [87].

### 4.5. Modern Imaging Methods

Modern imaging methods allow the precise localization of peripheral nerve damage. Sonoelastography [29] and ultrasonography investigations of the median nerve [88] are emerging techniques to image the median nerve. Researchers have used the ultrasonography investigation of the median nerve to visualize the median nerve in the carpal tunnel, revealing an increased median nerve cross-sectional area (CSA) and decreased echogenicity due to neural edema within the carpal tunnel [51]. Sonoelastography, previously applied to document decreased skin elasticity in patients with SSc, now is also used for median nerve imaging [51]. Yagci et al. [29] completed investigations of 47 hands in 24 SSC patients, 53 hands in 27 CTS patients and 38 hands of healthy controls, and found that the elastic ratio at the psiform and forearm levels in the SSc group were significantly higher than in the CTS and control groups, suggesting that the increased peripheral nerve involvement in SSc reflects increased stiffness of the nerves. Bignotti et al. [89] used high-resolution ultrasound with a computer-assisted assessment to quantitatively assess and compare the nerve densities in patients with limited cutaneous systemic sclerosis (lcSSc) and reported that nerve density was reduced in lcSSc patients. This was especially evident in the symptomatic group, confirming the role of ultrasound in improving the diagnostic accuracy of carpal tunnel syndrome. Anno et al. [90] performed ultrasound real-time tissue elastography to test 402 hands in 201 RA patients and 222 hands in 111 non-RA patients and found that nerve stiffness was higher in RA patients than in non-RA patients. They further predicted that an inflammatory condition of the flexor tendon and wrist joint in patients with RA may generate fibrotic changes in the median nerve.

Diffusion-weighted magnetic resonance neurography (DW-MRN) is another emerging technique that can exploit the greater water diffusion anisotropy in peripheral nerves for improved visualization [91]. Based on the concept of background body and vascular signal suppression for improved visualization of stationary fluid or cellular structures, DW-MRN is applicable to improved visualization of extremity nerves and their lesions in the wrist and palm with adequate image quality, thus providing a supplementary method for conventional magnetic resonance imaging [91].

## 5. New Treatments and Outcomes

### 5.1. Glucocorticoids with Immunosuppressants as Basic Therapy

There are no clear guidelines for the treatment of PN-CTD. In particular, the importance of PN when deciding the treatment strategy is often underestimated. In 2010, the guidelines of the Society for Neurology of USA recommended (with evidence level B) vasculitic peripheral neuropathy shock therapy with glucocorticoids, possibly in combination with immunosuppression as basic therapy [92]. Major treatment recommendations were: (1) corticosteroid (CS) monotherapy for at least 6 months as first-line treatment; (2) combination therapy applied to rapidly progressive and non-systemic vasculitic neuropathy and patients who progress on CS monotherapy; (3) immunosuppressive options including cyclophosphamide, azathioprine, and methotrexate; (4) cyclophosphamideapplies to severe neuropathies, generally administered in IV pulses to reduce cumulative doses and the associated side effects; (5) combination therapy for patients achieving clinical remission, followed by continued maintenance therapy for 18–24 months with azathioprine or methotrexate [92].

In 2019, Bortoluzzi et al. [7] systematically summarized the treatment regimen of PN-SLE. The treatment options encompassed corticosteroids, immunoglobulins and plasmapheresis, often accompanied by long-term immunosuppression [7]. In a multi-ethnic/racial, prospective SLE inception cohort with a mean follow-up of 7.6 ± 4.6 years by Hanly et al. [13] in 2020, the majority of neuropathies were resolved or improved over time during follow-up. The time to reach resolution was most rapid for cranial neuropathy, followed by mononeuropathy and polyneuropathy. In a retrospective study on PN-RA by Ding et al. [93], paresthesia in 27 patients was relieved after treatment with high dose glucocorticoids and immunoglobulins (IVIG). Twelve patients were followed up regularly with a mean duration of follow-up of 17.0 (4.8–52.8) months. Paresthesia in 10 (10/12) patients was relieved compared to those at discharge and 1 (1/12) patient achieved complete remission. 

Induction treatments with glucocorticoids with or without immunosuppressants was indicated for active vasculitic neuropathy [94], and therapy with rituximab, azathioprine or methotrexate was recommended for 2 years to maintain remission [5]. In the retrospective cohort study of Cho et al. [40], the long-term clinical outcome of PN-EGPA patients who received initial corticosteroid and cyclophosphamide combination therapy was favorable, producing a very low relapse rate.

### 5.2. Neuroactive Steroids in Neuropathic Pain

Glucocorticoids are routinely used to relieve pain. However, as reported in experimental models of neuropathic pain, glucocorticoids may contribute to hyperalgesia, exacerbate neuropathic pain, activate the early phase of pain induction and induce hyperalgesia [3]. Therefore, concerns have been raised about the use of glucocorticoids for pain treatment [3] and a new strategy is needed. Neuroactive steroids (i.e., steroid hormones synthesized by peripheral glands and those directly synthesized in the nervous system) represent critical physiological regulators of PNS function and can have protective impacts on several symptoms of PN including neuropathic pain [4]. Falvo et al. [4] summarized the role of neuroactive steroids for the treatment of neuropathic pain thus: progesterone prevents pain-related behaviors, such as allodynia and hyperalgesia, in different models of neuropathic pain; tetrahydroprogesterone reduces thermal and mechanical hyperalgesia by enhancing the activity of the γ-amino butyric acid-A receptor and by blocking T-type Ca^2+^ channels; testosterone exerted anti-nociceptive effects in neuropathic rats; 17β-estradiol caused pain attenuation and a decrease of neuropathy-induced gliosis after sciatic nerve ligature; moreover, it exerted protective effects on neuropathic pain via estrogen receptors by inhibiting microglia activation and the production of inflammatory mediators. These observations support the view that neuroactive steroids may provide an interesting topic for neuropathic pain research in the near future.

## 6. Conclusions

PN is a well-known consequence of CT, with a complex spectrum of overlapping clinical manifestations of rheumatologic and non-rheumatologic disorders, such as infections, drug toxicity or metabolic disease. Therefore, it is still a clinical issue and a challenge to rheumatologists and neurologists alike. The reported prevalence and clinical manifestations of PN in CTD varies widely. Recent studies have shown that seropositive RA and sSS may have a higher incidence of PN than seronegative RA and pSS. Furthermore, allergic inflammation was an underlying mechanism in MPO-ANCA negative EGPA patients. Although the majority of PN patients are resolved or improved over time by induction treatments, no reliable prediction of the treatment response or accepted diagnostic tools are available. Given the recent insights in PN-CTD pathogenesis and a better understanding of its disease course, new therapeutic targets and clinical management may be approved to improve the outcome of PN involvement in CTD in the future. However, some limitations of this review should also be considered, such as the wide variation in methods used to report complications and the frequency of reporting as determined by studies with different sample sizes in multiple age groups. Prospective clinical trials are needed for early detection of all possible medication-related side effects. Long-term evaluation of patients is also important to control recurrence.

Key messages: The reported prevalence and clinical manifestation of PN in CTD varies widely;The main mechanism of PN is different for each CTD and the pathogenesis may also vary in different subtypes of the same disease;Neuroactive steroids may have protective effects against several PN features, especially neuropathic pain.

## Figures and Tables

**Figure 2 diagnostics-11-01956-f002:**
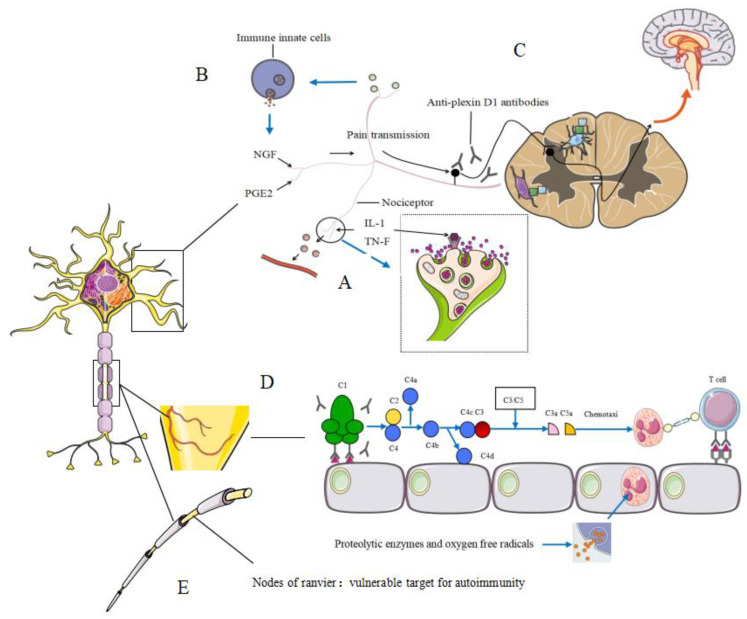
Pathogenesis of peripheral neuropathy and neuropathic pain in connective tissue disease. (**A**) I L-1β, IL-6 and TNF-α are sensed by nociceptors, which then leads to the release of various neuropeptides, such as calcitonin gene-related protein and substance P, resulting in vasodilatation and increased vascular permeability. (**B**) The mediators released from sensory neurons in the periphery attract and activate immune innate cells and adaptative immune cells, release NGF and PGE2, inducing peripheral sensitisation and hyperalgesic phenomena. (**C**) (1) The invasion of anti-plexin D1 antibodies in the dorsal root ganglia (DRG) on which the BBB and blood–nerve barrier are absent and binding to unmyelinated small DRG neurons (primary sensory neurons), cause neuropathic pain. (2) ET-1 (endothelin-1) causes BBB hyperpermeability and activates microglia and astroglia through the ET-1/EDNRB (endothelin receptor type B) pathway in allergic inflammation. Then glial activation triggers the activation of second-order sensory neurons in the dorsal horn of the spinal cord and induces neuropathic pain. (**D**) (1) Cell-mediated immunity: circulating T cells recognize antigens related to endothelial cells acting as antigen-presenting cells, Increased expression of cell adhesion molecules and release of chemotactic cytokines caused by this interaction, in turn, recruit and activate neutrophils and lymphocytes with subsequent inflammation and destruction of the vascular wall. (2) Deposition of immune complexes (IC): circulating antibodies binding to endogenous or exogenous antigens forms IC which deposits on the vascular wall and activates the complement system, then the factors C3a and C5a chemotactic for neutrophils are formed. Neutrophils could infiltrate the vascular wall, phagocyte the IC and release proteolytic enzymes and oxygen-free radicals that destroy the vascular wall. (**E**) Due to the substantially increased number of potential antigens and the crucial permeability of the blood–nerve barrier in nodal and juxta paranodal structures, nodes of Ranvier may be a vulnerable target for autoimmunity.

**Table 2 diagnostics-11-01956-t002:** Associated factors related to PN in patients with CTD.

Associated Factors
Systemic lupus erythematosus
Older age at SLE diagnosis [10,12,13]
Higher SLE Disease Activity Index 2000 scores [11,12,13,14]
Cutaneous vasculitis [12,14]
Sjögren’s syndrome
Salivary gland enlargement [22,26]
Hypocomplementemia [17,22]
Eosinophilic granulomatosis with polyangiitis
Myeloperoxidase-ANCA positivity [38,39]
Skin, musculoskeletal, cardiovascular involvement [38]
Rheumatoid arthritis
DAS-28 [49]
Inflammatory markers of disease activity [49,78]
Increased platelet numbers [78]

## Data Availability

Search results are available from the authors.

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
