# Peer review of "Clinical Manifestations, Pathogenesis, Diagnosis and Treatment of Peripheral Neuropathies in Connective Tissue Diseases: More Diverse and Frequent in Different Subtypes than Expected"

_diagnostics, 2021, doi:10.3390/diagnostics11111956_

Round 1

Reviewer 1 Report

it is necessary to talk about the diagnosis of small fibers neuropathy referring to qst, laser evoked potentials and skin biopsy and to expand new therapies part

Author Response

Point 1: it is necessary to talk about the diagnosis of small fibers neuropathy referring to qst, laser evoked potentials and skin biopsy and to expand new therapies part

Response 1: In lines 87 to 105 we added the definition of small fibers neuropathy. In lines 429 to 448, we added a section on diagnosis of small fibers neuropathy referring to qst, laser evoked potentials and skin biopsy, etc.  In the “New treatments and outcomes” section, we have made a summary of “Neuroactive steroids in Neuropathic pain”.  But, so far it has only been demonstrated in animal models, and hopefully it will be used in patients in the future with more clinical data available.

Reviewer 2 Report

Strength of manuscript:

1, the topic of this article is a very good and original idea – it belongs to one of the most underinvestigated areas of clinical immunology/rheumatology.

2, Data from the last 6-year were analysed: it is a relatively long assessment period.

3, it is a review about not only the clinical manifestations of peripheral neuropathy in different CTDs, but about pathogenesis, diagnosis and treatment too.

4, Most of the results are in accordance with our clinical (not evidence based) experience: the prevalence and the importance of peripheral neuropathy is very different in our patients with CTDs too.

Wantingness of manuscript:

1, The searching methods is not too detailed:

a, the authors used only Pubmed, the other well-known databases not.

b, there are no information, which types of publications were analyzed – only original articles or other types too, such as letters, congress presentations, etc.

            c, The search terms were very simple: only “peripheral neuropathy”, other possible phrases – nerve or neurological manifestations/symptoms/involvement or neuropsychiatric or nervous system, etc were not examined.

2, This review analyzed data only from 2015, without any summary of earlier informations in this topic, so we not know exactly from this review what is new or different compared with the previous experiences (there is a such short description in the treatment chapter but there is no any sketch about it in the others).

3, There are some interesting/debatable conclusions too.

a, For example to Sjögren’s syndrome: “In conclusion, PN events are uncommon in SS.” I think the prevalence of peripheral neuropathy of 19% to 72% is high.

b, to PAN: “Mononeuritis multiplex is the main form of PN manifestation, suggesting that the underlying mechanisms are arteritis of the vasa nervorum and neurogenic inflammation.” Why these two machanisms? Why not only one of them? It is possible to conclude any pathogenetic relevance from clinical informations?

Suggestions   

1, I suggest to use more databases and search terms.

2, Please write a brief summary about previous information in all chapter or give an up-to-date short outline (e.g. in a table) about evidences. I recommend to use the data of your figure 1 for all diseases. Sjögren sy.: axonopathy ++, myelinopathy ++, neuronopathy+; mononeuropathy +, mononeuropathy multiplex+, polyneuropathy ++; autonomous neuropathy?, etc. Or similar descriptions with other simple method, e.g. yes/no /?.

3, Please reconsider some of the conclusions.

Author Response

Point 1: The searching methods is not too detailed:

a, the authors used only Pubmed, the other well-known databases not.

Response 1a: We tried using PubMed, Embase and Cochrane to retrieve papers at first, and nearly 4000 papers needed to be retrieved after keywords search, which was a huge workload. However,  we found that most excellent papers could be retrieved in PubMed, so PubMed was our main retrieval method.  In this revision, we use Embase and Cochrane again for retrieval and added some related content about definition and diagnosis for small fibers neuropathy.

b, there are no information, which types of publications were analyzed – only original articles or other types too, such as letters, congress presentations, etc.

Response 1b:: We add a  sentence in the revision: A literature review of original articles, review articles and case reports was conducted using Pubmed, Embase and Cochrane databases from January 2015 to June 2021.(Line 54 to 55)

c, The search terms were very simple: only “peripheral neuropathy”, other possible phrases – nerve or neurological manifestations/symptoms/involvement or neuropsychiatric or nervous system, etc were not examined.

Response 1c:: That is a good suggestion. We tried using the phrases the reviewer suggested to search, however, no more papers with new or valuable information were found. Actually, we have searched enough numbers of articles about connective tissue disorders involving the nervous system, including the central and peripheral nervous systems, and summarized them by sifting out important information.

Point 2:  This review analyzed data only from 2015, without any summary of earlier informations in this topic, so we not know exactly from this review what is new or different compared with the previous experiences (there is a such short description in the treatment chapter but there is no any sketch about it in the others).

Response 2:  As a heterogeneous group of neurological disorders, the reported prevalence and clinical manifestations of PN in CTD vary widely.  So comparison with earlier information about the prevalence and clinical manifestations is somewhat difficult. Actually, we tried to compare some interesting results with previous studies in the manuscript. For example, “Several previous studies found a higher frequency of symmetric sensorimotor polyneuropathy and symmetric sensory polyneuropathy, but mononeuropathy or mononeuritis multiplex was the most common pattern in these studies”(Line 152-155).  Moreover, the novelty of our work is to summarize the differences of PN events among different subtypes of connective tissue diseases.

Point 3 There are some interesting/debatable conclusions too.

a, For example to Sjögren’s syndrome: “In conclusion, PN events are uncommon in SS.” I think the prevalence of peripheral neuropathy of 19% to 72% is high.

Response 3a: This is a clerical error and “uncommon” has been changed to “common”(Line 164).

b, to PAN: “Mononeuritis multiplex is the main form of PN manifestation, suggesting that the underlying mechanisms are arteritis of the vasa nervorum and neurogenic inflammation.” Why these two machanisms? Why not only one of them? It is possible to conclude any pathogenetic relevance from clinical informations?

Response 3a: The conclusion has been reworked to make it easier to understand and persuasive. (Line 191 to 194)

Reviewer 3 Report

Jin and Liu reviewed the clinical manifestation, pathogenesis, diagnosis, and innovative treatment of PN in CTD. The review appears comprehensive and well-structured. I think this study deserves to be published but I have some comments.

  • In the “prevalence and clinical manifestation” paragraph, authors should take in to account the time of onset (acute, subacute, chronic) of PN (eventually add also in the figure 1). Moreover, in this section authors have to cite the length- (involvement of lower limbs with sparing of upper limbs) or non-length-dependent (involvement of face, mouth, trunk and upper limbs) pattern of PN.
  • In lines 120-122, authors described a length-dependent pattern neuropathy.
  • In the paragraph 4.3, authors cited skin biopsy how a method able to discriminate length- from non-length-dependent neuropathy. Actually, it is possible when skin biopsy is performed from thigh and distal leg (doi:10.1111/ene.13608). Moreover, authors have to specify that small-fiber neuropathy is diagosed by a reduction of IENF density and/or abnormal quantitative sensory testing (doi:10.1111/j.1468-1331.2010.03023.x). Lastly, skin biopsy let to study, over the epidermal nerve fibers, also the autonomic fibers that innervate sweat gland, pilomotor muscle and hair follicles (doi:10.3390/brainsci10120989).
  • In 3.4 paragraph, authors cited anti-paranodal antibody (anti-NF155) as useful laboratory test. Actually, to date, no patients were reported to have a CTD and demyelinating neuropathy due to anti-paranodal antibody. I think that in a patient suffering from CTD, anti-paranodal antibody research is not required unless a clear demyelinating neuropathy is present.  
  • Some typos/errors are present: acupuncture (sharp) on line 89, compressional on line 257, demyelization on line 343. Check the manuscript.

Author Response

Point 1: In the “prevalence and clinical manifestation” paragraph, authors should take in to account the time of onset (acute, subacute, chronic) of PN (eventually add also in the figure 1). Moreover, in this section authors have to cite the length- (involvement of lower limbs with sparing of upper limbs) or non-length-dependent (involvement of face, mouth, trunk and upper limbs) pattern of PN.

Response 1:  This is a very meaningful suggestion, but unfortunately, when the original texts in the papers we have found were searched, the authors did not indicate the onset of the PN -CTD.

In line 92 to 105, we add a description of “length-dependent” SFN and “non-length-dependent” SFN.

Point 2: In the paragraph 4.3, authors cited skin biopsy how a method able to discriminate length- from non-length-dependent neuropathy. Actually, it is possible when skin biopsy is performed from thigh and distal leg (doi:10.1111/ene.13608). Moreover, authors have to specify that small-fiber neuropathy is diagosed by a reduction of IENF density and/or abnormal quantitative sensory testing (doi:10.1111/j.1468-1331.2010.03023.x). Lastly, skin biopsy let to study, over the epidermal nerve fibers, also the autonomic fibers that innervate sweat gland, pilomotor muscle and hair follicles (doi:10.3390/brainsci10120989).

Response 2: These suggestions have been supplemented in the “Histopathological techniques” section.

Point 3: In 3.4 paragraph, authors cited anti-paranodal antibody (anti-NF155) as useful laboratory test. Actually, to date, no patients were reported to have a CTD and demyelinating neuropathy due to anti-paranodal antibody. I think that in a patient suffering from CTD, anti-paranodal antibody research is not required unless a clear demyelinating neuropathy is present.  

Response 3: The purpose of quoting NF155 here is only to extend the concept of IgG4-mediated autoimmune diseases, so as to better demonstrate that these immune antibody markers are useful in clinical practice. We agree with the reviewer’s opinion that anti-NF155 test is required only in patients with a clear demyelinating neuropathy, not in all CTD patients.

Point 4:Some typos/errors are present: acupuncture (sharp) on line 89, compressional on line 257, demyelization on line 343. Check the manuscript.

Response 4: After checking the manuscript carefully, these mistakes, along with others, have been corrected.

Round 2

Reviewer 2 Report

Thank you very much for your responses